# Resolving Interference (*RI*): Disentangling Models for Improved Model Merging

## Abstract

Model merging has shown that multitask models can be created by directly combining the parameters of different models that are each specialized on tasks of interest. However, models trained independently on distinct tasks often exhibit interference that degrades the merged model's performance. To solve this problem, we formally define the notion of **Cross-Task Interference** as the drift in the representation of the merged model to its constituent models. Reducing cross-task interference is the key to improving merging performance. To address this issue, we propose our method **Resolving Interference (RI)**, a light-weight framework which disentangles expert models to be functionally orthogonal to the space of other tasks, thereby reducing cross-task interference. RI does this whilst using only *unlabeled auxiliary* data as input (i.e., no task-data is needed), allowing it to be applied to under data-scarce scenarios. RI consistently improves the performance of existing merging methods by up to **10%** and generalization to unseen domains by up to **2.3%**. We also find RI to be robust to the source of auxiliary input while being significantly less sensitive to tuning of merging hyperparameters.

## 1 Introduction

Model merging has achieved remarkable success in recent years, showing that multitask models can be constructed by directly combining the parameters of independently trained specialist models (Wortsman et al., 2022; Choshen et al., 2022; Matena & Raffel, 2022; Stoica et al., 2024a;b; Yadav et al., 2023; Ilharco et al., 2023). These models can retain the specializations of their constituent models (Wortsman et al., 2022; Choshen et al., 2022; Matena & Raffel, 2022; Ilharco et al., 2023; Stoica et al., 2024a), integrate their skills to solve new tasks (Stoica et al., 2024a;b), or even improve out-of-distribution robustness beyond the original models (Rame et al., 2022).

However, obtaining these strong merged models is still challenging. Merging quality is dependent on the parameters' compatibility, the distributions used for pretraining and fine-tuning, and—most critically—the degree of conflicting parameters between the models being merged (Hammoud et al., 2024; Stoica et al., 2024b; Wang et al., 2024a; Yadav et al., 2023; Matena & Raffel, 2022; Stoica et al., 2024b; Yu et al., 2024). These conflicts lead the merged model's output representations to drift away from those of its constituent models—a phenomenon known as "cross-

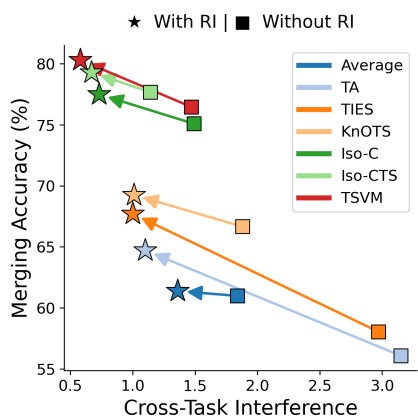

Figure 1: **Resolving Interference (RI)** is a lightweight adaptation strategy that mitigates cross-task interference, enhancing the performance of existing model-merging techniques.

task interference"—and degrade its performance. Existing methods that attempt to reduce interference fall within two broad categories: gradient-free and gradient-based. Gradient-free approaches (e.g., Yadav et al., 2023; Yu et al., 2024; Stoica et al., 2024b) are the most common merging methods, and typically try to reduce interference by operating directly on the parameters themselves in the absence of data. However, ignoring the data inherently limits these approaches, preventing gradient-free methods from capturing how model parameters interact with inputs. In contrast, gradient-based

approaches might alleviate this through optimization and assumed access to data, however, their objectives are not explicitly designed to reduce interference (Yang et al., 2024b; Ortiz-Jimenez et al., 2024). Importantly, these methods rely on the availability of the *original data distributions* used to train each constituent model incorporated in merging, which may not be realistic in practice, especially when the data is scarce/proprietary. Gradient-free merging approaches also tend to be sensitive to merging hyperparameters which are again tuned on task-specific validation data. There's also a third category of methods, which involves a mix of merging and routing (Dhasade et al., 2025; Lu et al., 2024; Yang et al., 2024a), that can mitigate interference better with extra capacity but incur additional inference cost and task-specific routing.

We ask an existential question: is there a *lightweight* gradient-based approach that can help reduce cross-task interference without increasing inference cost, and—*crucially*—*without requiring access* to the data distributions of the constituent models? We address this challenge by (1) formally defining the notion of cross-task interference, and (2) proposing an adaptation framework that minimizes this objective with just *arbitrary* auxiliary data, —yielding stronger merged models. Whereas prior work has proposed definitions for quantifying interference, they do not explicitly optimize towards reducing it (Ilharco et al., 2023; Yadav et al., 2023; Yu et al., 2024; Ortiz-Jimenez et al., 2024; Tang et al., 2024; Stoica et al., 2024b). In contrast, our formulation reveals how parameters can be adapted to reduce interference even before they are merged.

Building on this insight, we introduce *Resolving Interference (RI)*, a lightweight adaptation framework that enforces expert models into disjoint functional subspaces, thereby reducing cross-task interference. RI does this using non-task-specific auxiliary data, after which existing merging methods can be applied. Notably, RI improves merging performance over state-of-the-art merging methods by up to *10%* across a variety of model scales and challenging benchmarks. Moreover, we show that RI-merged models enhance out-of-distribution robustness by up to *2.3%* in the difficult DomainNet benchmark. We further dissect each component of our method through extensive ablations.

Our contributions are summarized as follows: (1) **Interference formalization**: We introduce a metric that captures *cross-task interference*, and whose reduction meaningfully minimizes parameter conflicts between models while simultaneously providing a principled diagnostic for merge quality. (2) **Framework**: We propose *Resolving Interference (RI)*, a framework that efficiently reduces this objective using *auxiliary* data. (3) **Empirical results**: We show that RI consistently improves merging performance by up to 10% across diverse benchmarks and model scales, and further enhances out-of-distribution robustness by up to 2.3%. (4) **Analysis**: We perform extensive ablations over data sources and optimization strategies, offering practical recommendations for applying RI effectively.

## 2 Related Works on Merging Interference

**Model Merging.** Model merging seeks to integrate independently trained models into a single unified model. Early studies revealed that models trained from the same initialization can be linearly interpolated without increasing test error—a phenomenon known as *mode connectivity* (Draxler et al., 2018; Garipov et al., 2018; Simsek et al., 2021; Frankle et al., 2020; Neyshabur et al., 2020). Subsequent work demonstrated that simple weight averaging not only preserves accuracy but can also improve generalization and reduces overfitting (Choshen et al., 2022; Wortsman et al., 2022; Wang et al., 2024a; McMahan et al., 2017; Rame et al., 2024; 2022). Building on these findings, Ilharco et al. (2023) introduced the concept of a *task vector*—the difference between fine-tuned and pretrained weights that captures task-specific knowledge. Task vectors can be merged and then added back to the pretrained weight to produce a unified model, while keeping the pretrained backbone untouched to preserve its original capabilities. This task-vector paradigm has since gained widespread traction and inspired a range of more advanced merging techniques (Yadav et al., 2023; Yu et al., 2024; Stoica et al., 2024b; Tam et al., 2023; Wang et al., 2024c;b).

**Reducing Interference.** A key challenge in merging models trained on distinct tasks is *cross-task interference* (Ortiz-Jimenez et al., 2024), where the output representations of the merged model drift away from those of the constituent experts. To address this problem, a variety of *gradient-free*, *gradient-based*, and *routing-based* adaptation strategies have been proposed. Gradient-free methods attribute interference to noisy, low-magnitude gradient updates and mitigate it by pruning such updates (Yadav et al., 2023; Yu et al., 2024; Sun et al., 2025), or by explicitly aligning task vectors (Stoica et al., 2024b; Gargiulo et al., 2025; Marczak et al., 2025; Choi et al., 2024). Gradient-based approaches

include Fisher-weighted averaging (Matena & Raffel, 2022) and the learning of redundant vectors that, when subtracted from other task vectors, reduce interference (Xiong et al., 2024); These methods typically assume access to task-specific data and leverage it during adaptation. Other gradient-based techniques, such as Adamerging (Yang et al., 2024b), learn task- or layer-specific scaling coefficients and complement other interference-reduction methods.

Prior gradient-based approaches assume access to task-specific data, which may not be available due to privacy reasons or when operating in data-scarce scenarios. To address this problem, our work proposes a lightweight adaptation strategy that operates with any task-agnostic, unlabeled, auxiliary data to adapt expert models into orthogonal subspaces. Our method is not a merging method by itself but rather an adaptation framework which enhances the performance of other existing merging techniques.

## 3   PROBLEM SETUP

**Problem Setting**. We assume access to a collection of finetuned expert models, that specialize in distinct tasks (e.g., classifying breeds of dogs; breeds of cats). Each model is composed of: (1) a backbone and (2) a head specific to the task (i.e., "task-head"). Heads can be as simple as a linear-layer or complex neural networks. We assume that all models share the same backbone architecture and initialization, whereas their respective heads are custom-built for each task. Our goal is to merge these backbones into a single one capable of being paired with *any* task-head and solve its respective task. We operate in a data-scarce setting, where we *do not* have access to the data distributions from *any* tasks, not even validation data.

**Notation.** We assume access to a collection of $N$ finetuned *expert models* and denote each model's backbone by $f(x|\theta)$ where $x$ is a sample from some data distribution, and $\theta$ parameterizes $f$. Additionally, we denote a task-head as a function $h$, and the model finetuned on the $i^{th}$ task using dataset $D_i \sim P_i$ as $h_i(f(x|\theta_i))$. Altogether, the set of our models is given by, $\left\{ h_i(f(\cdot|\theta_i)) \right\}_{i=1}^{N}$. Let $\theta_0$ be the shared initialization. Following Ilharco et al. (2023), we separate the finetuning specialization of model-$i$ on task-$i$ from $\theta_0$ into a "task-vector": $\tau_i = \theta_i - \theta_0$. Given the set of all task-vectors from all models, $\{\tau_1, \ldots, \tau_N\}$, a merging method $M$ produces a single "merged-vector": $\tau_m = M(\tau_1, \ldots, \tau_N)$. We then obtain the merged backbone with the following operation: $\theta_m = \theta_0 + \tau_m$. For all task-evaluations, we couple $\theta_m$ with the respective task head $h_i$: $h_i(f(x|\theta_m))$.

## 4   CROSS-TASK INTERFERENCE

Directly merging models that are finetuned on distinct tasks often results in significant performance degradation when evaluating the merged model on each constituent task (Yadav et al., 2023; Yu et al., 2024; Ortiz-Jimenez et al., 2024; Wang et al., 2024a; Stoica et al., 2024b;a). This conflict occurs because the learned features in one model may overwrite the features in a second when combined. For example, suppose we would like to merge a model that classifies different breeds of cats with one that specializes in distinguishing between different breeds of dogs. Ideally, the merged model should be able to identify cat or dog breeds with the same efficacy as its constituent models. However, the parameter-values representing the features useful for classifying cats may be distorted by those necessary for differentiating dogs, resulting in a merged model whose output representations fail to classify either correctly. This interference between features is known as "cross-task interference", and mitigating it is of paramount importance for successful merging (Stoica et al., 2024a; Ortiz-Jimenez et al., 2024). Developing a quantitative measure for this interference is therefore crucial for evaluating mitigation techniques by work to date.

**Cross-Task Interference** ($\xi$). We propose to measure cross-task interference as the deviation between the representations of the merged model and each task-expert model, when evaluated on its respective task:

$$\xi(\{\theta_i\}_{i=1}^{N}, \theta_m) = \sum_{i=1}^{N} \mathbb{E}_{x \sim P_i}[\text{dist}\left(h_i(f(x|\theta_i)), h_i(f(x|\theta_m))\right)], \tag{1}$$

where "dist" is a distance metric that quantifies the output representation difference between the merged model and the respective individual model. Depending on the setting, "dist" can take

many forms (e.g., KL-Divergence in classification to compare the estimated probability distributions between models). Note that $\xi = 0$ is a sufficient condition to guaranteeing that a merged model performs equally well to each of its constituent models, when evaluated on their respective tasks.

**Relation to other interference metrics.** $\xi$ may appear similar to the "disentanglement-error" introduced by (Ortiz-Jimenez et al., 2024), which measures how *scaling* the parameters of each task-expert affects the representations of the merged model. However, our definition of interference explicitly compares the representations of the merged model to those of the *original* task-experts.

## 5 Resolving Interference (RI)

One way to ensure that the merged backbone $\theta_m = \theta_0 + \tau_m$ minimizes $\xi$ is by making sure each of the constituent task vectors $\tau_i$ incorporated into $\tau_m$ are *functionally orthogonal* to the heads of other tasks. Specifically, each $\tau_i$ can be adapted to a $\tau_i^*$, where (1) $\tau_i^*$ only influences the output representations across $h_i$ when evaluated on $D_i$, and (2) $\tau_i^*$ has no influence when evaluated across the heads and data of other tasks. This functionality can be achieved by enforcing the following constraints:

$$h(f(x|\theta_0 + \tau_i^*) = \begin{cases} h(f(x|\theta_0 + \tau_i)), & x \in D_i, h = h_i \\ h(f(x|\theta_0)), & x \in D_j, h = h_j, \forall j \neq i \end{cases} \tag{2}$$

Thus, $\tau_i^*$ mimics $\tau_i$ under its task-head and data: $h_i(f(x|\theta_0 + \tau_i^*) = h_i(f(x|\theta_0 + \tau_i))$ when $x \in D_i$. Similarly, $h_j(f(x|\theta_0 + \tau_i^*)) = h_j(f(x|\theta_0))$ when $x \in D_j, \forall j \neq i$ ensures that $\tau_i^*$ bears no influence on the output representations of the other task-heads.

However, we cannot directly solve Eq. 2 because *we do not* assume access to *any* task-data $D_i$ in our setting. Thus, we propose to alter it to instead be defined over any[1] accessible auxiliary data $x \in D_{\text{aux}}$:

$$h(f(x|\theta_0 + \tau_i^*) = \begin{cases} h(f(x|\theta_0 + \tau_i)), & x \in D_{\text{aux}}, \quad h = h_i \\ h(f(x|\theta_0)), & x \in D_{\text{aux}}, h = h_j, \forall j \neq i \end{cases} \tag{3}$$

While constraining on $D_{\text{aux}}$ no longer guarantees that $\tau_i^*$ is a solution to Eq. 2, we find it to be a sufficient condition for our setting, and we empirically observe that enforcing it dramatically reduces $\xi$. In practice, we solve for Eq. 3 by optimizing the following loss objective which we define as the **Resolving Interference Loss or simply RI Loss**,

$$\mathcal{L}(x \in D_{\text{aux}}, \theta_0, \tau_i, \tau_i^*) = \text{dist}[h_i(f(x|\theta_0 + \tau_i)), h_i(f(x|\theta_0 + \tau_i^*))] +$$

$$\frac{\alpha}{N-1} \sum_{j=1, j \neq i}^{N} \text{dist}[h_j(f(x|\theta_0)), h_j(f(x|\theta_0 + \tau_i^*))] \tag{4}$$

It is trivial to observe that achieving dist $= 0$ in the red component of Eq. 4 satisfies the red component constraint in Eq. 3, and similarly achieving dist $= 0$ in the blue component of Eq. 4 satisfies the blue component of Eq. 3. Here, $\alpha$ controls the trade-off between preserving the original task behavior versus minimizing interference with other tasks. Using $\alpha = 1$ by default ensures equal importance to both objectives. To make $\alpha$ less sensitive to the number of tasks, we normalize the second term in Eq. 4 by dividing it by $N - 1$. Once all task vectors have been disentangled with RI with respect to the heads of other tasks, they can then be merged using standard merging techniques.

## 6 Experimental Setup

**Models.** We make use of CLIP models of different sizes, such as ViT-B/32, ViT-B/16 and ViT-L/14 vision encoders across various classification tasks. The task-specific heads are constructed by concatenating text embeddings obtained from CLIP's frozen text encoder after processing the class labels of each task. The predicted logits are obtained by a dot product between the embedding from the vision encoder and the class-label embeddings. Following prior work (Gargiulo et al., 2025; Marczak et al., 2025), we make use of model checkpoints across 20 tasks from Wang et al. (2024c).

**Merging Baselines.** We evaluate our adaptation technique RI along with other prominent merging techniques such as Weight Averaging (Choshen et al., 2022; Wortsman et al., 2022), Task-Arithmetic

---

[1] We ablate different data choices in Section 8.

Example Setup: Consider merging (1) Dog Classifier $f(\theta_0 + \tau_{dog})$ and (2) Cat Classifier $f(\theta_0 + \tau_{cat})$

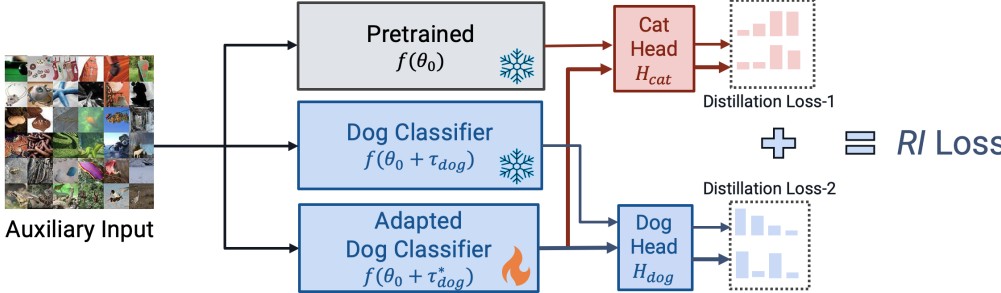

Figure 2: **Resolving Interference (RI)** for the dog classifier involves passing unlabeled auxiliary images through (i) the frozen pretrained backbone $f(\theta_0)$, (ii) the frozen dog classifier $f(\theta_0 + \tau_{\text{dog}})$, and (iii) a trainable copy $f(\theta_0 + \tau_{\text{dog}}^*)$. A twin distillation loss preserves output distribution across the dog head $h_{\text{dog}}$ while forcing output across all other heads (e.g., $h_{\text{cat}}$) to match the pretrained backbone, producing an adapted task vector $\tau_{\text{dog}}^*$. The same adaptation strategy is repeated for all expert models.

(Ilharco et al., 2023), TIES (Yadav et al., 2023), KnOTS-TIES (Stoica et al., 2024b), TSV-M (Gargiulo et al., 2025), Iso-C (Marczak et al., 2025) and Iso-CTS (Marczak et al., 2025). For simplicity, we refer to 'KnOTS-TIES' as just 'KnOTS' throughout this paper. The last few years have seen an overwhelming number of merging techniques being introduced (Wang et al., 2024c; Matena & Raffel, 2022; Jin et al., 2022; Wang et al., 2024b; Tam et al., 2023; Yu et al., 2024). Although the list of merging methods we test against is not exhaustive, we believe it covers a mix of prominent and current state-of-the-art merging techniques. Since our focus in this work is on the data-scarce setting, where we assume no access to task data, we regard all task-data-based adaptation (Yang et al., 2024b) and techniques which require additional capacity or task-specific indexing, such as routing-based strategies (Lu et al., 2024; Yang et al., 2024a), to be out of scope.

**Merging Hyperparameters.** We focus on the data-scarce setting, where access to task-specific data is entirely unavailable—even for validation. In such cases, we adopt the default merging hyperparameter values recommended in the original works. The list of hyperparameters include scaling coefficient, $top\text{-}k$ pruning factor and fraction of common-space. We list the hyperparameters used across different merging methods, depending on the number of tasks being merged, in the Appendix. We later discuss in Section.8 how models adapted with RI are significantly less sensitive to tuning.

**Resolving Interference (RI).** In order to optimize over the RI loss (Eq.4) we choose KL-divergence as the distance metric. We analyze a bunch of other metrics choices in Sec.8 and find KL-Divergence to be the most effective. For the source of auxiliary data we use unlabeled images from the ImageNet (Deng et al., 2009) dataset. We find setting the hyperparameter $\alpha$ in Eq. 4 to its default value 1.0 leads to stable interference resolution. During RI, we use a learning rate of $1e{-}6$ and a weight decay of $1e{-}4$. We stick to this configuration across all our experiments though one may tune them to optimise over the cross-task interference metric in Eq. 4. Further, to make our method lightweight, we apply RI for just 2500 training steps across a batch size of 128 for ViT-B/32 and ViT-B/16 models and use a reduced batch size of 32 for ViT-L/14 due to memory constraints. We run all our experiments on Nvidia A40 GPU with 48GB VRAM, which takes about 15 minutes to adapt each expert in the ViT-B/32, 8 task vision setting described in Sec. 7.1.

## 7 RESULTS

We evaluate the effectiveness of reducing interference using our lightweight adaptation strategy RI. Specifically, we merge expert models with a variety of prominent merging techniques and then assess the multitasking capabilities of the resulting merged model by measuring its performance on each task individually. Our evaluation covers vision benchmarks designed to probe both **in-domain accuracy** and **out-of-domain generalization** on DomainNet, allowing us to rigorously test the robustness of interference reduction.

| Merging Method | ViT-B/32 | | | ViT-B/16 | | | ViT-L/14 | | |
|---|---|---|---|---|---|---|---|---|---|
| | 8 tasks | 14 tasks | 20 tasks | 8 tasks | 14 tasks | 20 tasks | 8 tasks | 14 tasks | 20 tasks |
| Zero-shot | 48.3 | 57.2 | 56.1 | 55.3 | 61.3 | 59.7 | 64.7 | 68.2 | 65.2 |
| Finetuned | 92.8 | 90.9 | 91.3 | 94.6 | 92.8 | 93.2 | 95.8 | 94.3 | 94.7 |
| Averaging | **66.1** | 64.3 | 61.0 | **72.3** | **69.4** | **65.3** | **79.5** | **76.7** | **71.6** |
| Averaging + RI (Ours) | 65.7 | **64.6** | **61.4** | 71.3 | 69.1 | 64.9 | 78.9 | 75.8 | 71.1 |
| TA | 69.2 | 60.7 | 56.1 | 75.2 | 67.3 | 63.1 | 84.8 | 78.8 | 73.3 |
| TA + RI (Ours) | **76.5** | **70.3** | **64.7** | **81.4** | **75.9** | **69.5** | **87.3** | **82.1** | **76.4** |
| TIES | 73.7 | 64.7 | 58.0 | 79.5 | 71.2 | 65.8 | 86.7 | 78.7 | 74.3 |
| TIES + RI (Ours) | **79.3** | **72.8** | **67.7** | **84.1** | **78.4** | **72.9** | **89.4** | **82.6** | **79.1** |
| 78.7KnOTS | 77.2 | 72.1 | 66.7 | 81.7 | 77.2 | 71.1 | 87.8 | 81.1 | 78.8 |
| KnOTS + RI (Ours) | **78.4** | **74.0** | **69.3** | **83.1** | **78.4** | **73.0** | **88.4** | **83.1** | **79.7** |
| Iso-C | 86.3 | 79.9 | 75.1 | 90.3 | 84.5 | 79.5 | 94.2 | 89.5 | 87.7 |
| Iso-C + RI (Ours) | **87.0** | **81.4** | **77.5** | **90.8** | **85.9** | **82.0** | **94.3** | **89.9** | **88.6** |
| Iso-CTS | 86.6 | 81.6 | 77.7 | 91.0 | 86.2 | 82.2 | 94.7 | 91.0 | 90.0 |
| Iso-CTS + RI (Ours) | **86.8** | **82.5** | **79.3** | **91.1** | **87.0** | **83.6** | **94.7** | **91.0** | **90.2** |
| TSV-M | 85.4 | 79.9 | 76.5 | 88.7 | 84.4 | 80.4 | 92.8 | 89.1 | 87.7 |
| TSV-M + RI (Ours) | **87.0** | **82.7** | **80.3** | **90.0** | **86.0** | **83.2** | **93.5** | **90.2** | **89.3** |

Table 1: **Resolving Interference (RI) consistently improves existing merging techniques** on the popular 8/14/20 task vision benchmarks. Achieving upto 10% improvement on TA and pushing SOTA merging techniques like TSVM by up to 3.8 %. Each entry reports average accuracy. In line with our focus on the **task-data–free setting** where tuning merging hyperparameters is not possible, we make use of the **recommended defaults** for all merging baselines with and without RI.

## 7.1 MERGING 8/14/20 VISION TASKS

We use the image classification benchmark introduced by (Wang et al., 2024c), which evaluates the performance of the merged models across 8/14/20 vision datasets (full list in Appendix). For the task experts, we use model checkpoints from (Wang et al., 2024c).

In Table 1, we report the average accuracy of the merged model with and without RI across ViT-B/32, ViT-B/16 and ViT-L/14 vision transformers. We find RI consistently improves merging methods across the 8/14/20 tasks and across the different model architectures. Notably, we find methods such as Task-Aritimentic (TA) and TIES improve performance by (7.4%, 9.6%, 8.6%) and (5.6%, 8.1%, 9.7%) on ViT-B/32 based (8/14/20) task setting. We also observe significant gains across the state-of-the-art merging methods such as KnOTS (+2.6%), Iso-C (+2.4%), Iso-CTS (+1.6%) and TSV-M (+3.9%) on the 20-task ViT-B/32 setting. Since cross-task interference becomes a bigger issue with increasing number of tasks, resolving interference (RI) is even more effective at scale. Notably, we observe increased gains by (+1.5, +2.8, +3.9) with TSV-M and (+0.2, +0.9, +1.6) with Iso-CTS on the ViT-B/32 (8/14/20) tasks. While improved pre-training and model size are known to alleviate cross-task interference, we continue to observe consistent gains in performance even across the ViT-B/16 and the larger ViT-L/14 models. The only method which observes little to no gain in performance is simple averaging. This is due to the scale associated with averaging being too low. The TA baseline, which uses a higher scaling coefficient in general, results in higher baseline performance and observes a significant boost in performance with the addition of RI.

We analyze this case later in section and find the low scaling co-efficient associated with averaging to not be ideal to observe gains with RI.

## 7.2 OUT-OF-DOMAIN GENERALIZATION ON DOMAINNET

We evaluate the ability of the merged model to generalize to unseen distributions using the DomainNet dataset, which comprises images across 345 classes of common objects across 6 domains/image-styles: real, clipart, infograph, painting, quickdraw, and sketch. We partition each domain into two subsets—Split-0 containing the first 172 classes and Split-1 containing the remaining 173 classes. On

| Merging Method | Clipart | | Infograph | | Painting | | Quickdraw | | Sketch | | Mean(%) |
|---|---|---|---|---|---|---|---|---|---|---|---|
| | S-0 | S-1 | S-0 | S-1 | S-0 | S-1 | S-0 | S-1 | S-0 | S-1 | |
| Split-0 model | **76.2** | 70.0 | **49.3** | 38.0 | **67.7** | 51.9 | **17.2** | **18.5** | **66.3** | 59.4 | 51.5 |
| Split-1 model | 70.1 | **73.9** | 44.7 | **38.4** | 64.7 | **55.6** | 13.3 | 17.9 | 62.0 | **61.1** | 50.2 |
| Averaging | 76.5 | 74.7 | 50.1 | 40.1 | 69.2 | 54.9 | 16.1 | 20.0 | 67.7 | 62.7 | 53.2 |
| Averaging + RI (Ours) | **79.1** | **76.3** | **51.9** | **46.7** | **70.7** | **56.2** | **18.0** | 19.7 | **69.6** | **64.1** | **55.2** |
| TA | 77.5 | 75.6 | 51.0 | 41.0 | 70.1 | 55.5 | 16.9 | **20.3** | 68.5 | 63.6 | 54.0 |
| TA + RI (Ours) | **79.1** | **75.9** | **51.9** | **41.8** | **70.8** | **56.1** | **18.2** | 19.6 | **69.8** | **64.0** | **54.7** |
| TIES | 75.5 | 73.5 | 48.6 | 38.8 | 67.4 | 54.0 | 15.0 | 18.9 | 66.1 | 61.5 | 51.9 |
| TIES + RI (Ours) | **78.4** | **76.3** | **51.8** | **40.7** | **69.7** | **56.0** | **17.4** | **19.3** | **69.0** | **63.7** | **54.2** |
| KnOTS | 75.8 | 74.6 | 48.8 | 39.5 | 69.0 | 54.8 | 15.0 | 19.2 | 66.5 | 61.9 | 52.5 |
| KnOTS + RI (Ours) | **78.8** | **76.1** | **51.3** | **41.4** | **71.2** | **55.8** | **17.1** | **19.6** | **69.3** | **63.6** | **54.4** |
| Iso-C | **79.0** | **76.3** | 51.1 | 41.0 | 71.4 | **56.3** | 17.4 | **20.4** | 69.3 | 63.5 | **54.6** |
| Iso-C + RI (Ours) | **79.0** | 75.9 | **51.6** | 41.0 | **71.6** | 56.2 | **17.7** | 19.8 | **69.5** | **63.7** | **54.6** |
| Iso-CTS | **78.8** | **76.1** | 51.2 | **40.8** | 71.0 | **56.0** | **17.7** | **20.4** | 68.9 | **63.5** | **54.5** |
| Iso-CTS + RI (Ours) | **78.8** | 75.6 | **51.6** | 40.7 | **71.1** | **56.0** | 17.6 | 19.5 | **69.4** | 63.3 | 54.4 |
| TSV-M | 76.4 | 74.3 | 49.6 | 39.5 | 68.4 | 54.8 | 16.0 | **19.5** | 67.1 | 62.3 | 52.8 |
| TSV-M + RI (Ours) | **79.0** | **76.1** | **51.9** | **41.0** | **70.8** | **56.2** | **17.3** | 19.4 | **69.2** | **63.7** | **54.5** |

Table 2: **Resolving Interference (RI) improves the generalization** performance of merging techniques on unseen domains by upto 2.3%, while improving by up to 3.9% compared to the task-expert models. Each dataset is split into two parts (denoted Split-0 (S-0) and Split-1 (S-1)), and the performance of original split-specific experts is reported in the top grey rows.

each split, an independent CLIP-based ViT-B/32 model is fine-tuned on the *real* domain. We then test how well the merged model generalizes to the 5 unseen domains.

In Table 2, we compare the performance of the merged models—with and without RI —to the individual expert models trained on each split. First, we observe that merging baselines even without RI consistently outperform both the Split-0 and Split-1 expert models, underscoring the enhanced generalization capability achieved via merging. Resolving interference (RI) before merging further improves generalization performance even further. In particular, we observe that RI improves the performance of Averaging and TIES by +2.0% and +2.3%, respectively, on average across all the unseen domains.

These results suggest that while merging task-vectors across different tasks enhances generalization, it still suffers from cross-task interference. We hypothesize that while RI reduces interference to an extent by encouraging orthogonality among task-vectors, while the remaining shared representation space facilitates a positive transfer, leading to improved out-of-distribution performance.

## 8 ABLATIONS & ANALYSIS

In this section, we provide a comprehensive analysis of RI, examining its impact on disentanglement error and investigating different distance metrics for optimizing the RI loss. We further explore an alternative strategy for reducing cross-task interference error through multitask distillation on auxiliary data and analyze what characteristics make a dataset a strong source of auxiliary input for RI. Finally, we assess the sensitivity of RI to various hyperparameter tuning objectives.

To analyze our method at scale, unless otherwise specified, all experiments are conducted on the challenging **20-vision-task setup** introduced in Sec. 7.1 using ViT-B/32–based expert models.

**Does reducing the RI loss on auxiliary data reduce cross-task interference on task data?** We investigate this by optimizing each expert model with the RI loss for 25,000 steps. As shown in Fig. 3, the first 1,000 steps yield a steep decrease in the RI loss on auxiliary data, which corresponds to a sharp reduction in the cross-task interference of the merged model on task data and leads to improved merging performance. The objective begins to saturate after roughly 2500 steps, providing only marginal gains thereafter. To keep RI lightweight, we therefore apply it for only **2,500 steps**, which represents

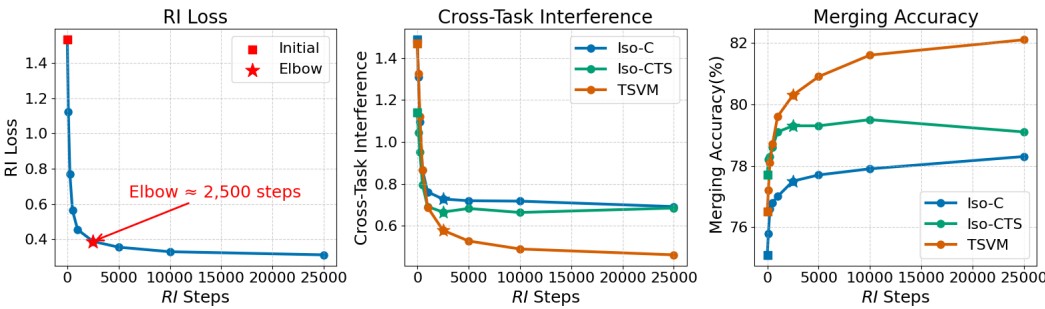

Figure 3: The Twin-Distillation loss (Eq.4) on auxiliary data reduces sharply (left), followed by a similar decline in the cross-task interference of the merged model measured on task-specific data (middle), which leads to significant improvement when using existing merging techniques(right).

the "elbow point" in Fig. 3 (left). Hence, we stick to adapting for 2500 steps in all our experiments. Though if computational resources permit, applying RI for longer can yield additional gains, we observe methods like TSV-M gain an additional 1.8%, reaching 82.1% when adapted for 25,000 steps.

**Which is the best metric to reduce the RI objective?** We explore different choices such as Mean-Square-Error (MSE), Cross-Entropy and KL-Divergence. As seen in the Table. 3 we observe that all three choices result in improved merging performance, where the KL-divergence based distance measure leads to an average improvement of 2.6% across various state-of-the-art merging methods.

**Can we directly adapt a single model using multitask distillation on auxiliary data?** We explore an alternative strategy for adapting a pretrained or merged model by applying the cross-task interference objective from Eq. 1 to auxiliary data. Building on the findings of Frank & Davis (2025), which show that *single*-task distillation can sometimes be performed using task-agnostic auxiliary data, we extend this idea to a more challenging setting: distilling knowl-

| Distance Metric | TSV-M | Iso-C | Iso-CTS | Average |
|---|---|---|---|---|
| **None (Without RI)** | 76.5 | 75.1 | 77.7 | 76.4 |
| **MSE** | 78.2 | 76.7 | 78.4 | 77.7 |
| RI **Cross-Entropy** | 79.9 | 76.3 | 78.5 | 78.2 |
| **KL-Divergence** | **80.3** | **77.5** | **79.3** | **79.0** |

Table 3: **Comparison of distance metrics** used to reduce RI loss on the merging performance. Minimizing *KL-Divergence* yields the highest improvement across merging methods.

edge from *multiple* tasks into a merged model using only unlabeled auxiliary data. We term this approach Merge+Distill$_{Aux}$. As reported in Table 8, this method produces only marginal gains over the standard merging baselines and is significantly lower compared to our Merge+RI approach. These results indicate that, while distillation with auxiliary data can reduce interference, it remains far less effective at learning multiple tasks simultaneously.

**What makes a good dataset to resolve interference?** To investigate this question, we consider the 8-vision-task setting described in Section 7.1 and individually evaluate datasets 9–14—namely CIFAR100, STL10, Flowers102, OxfordPets, PCAM, and FER2013—as sources of auxiliary data. In addition to the default ImageNet auxiliary dataset, we also test whether interference can be reduced by using purely synthetic input in the form of Gaussian noise. As shown in Fig. 4, datasets containing high-resolution, full-color images—such as ImageNet, OxfordPets, Flowers102, and STL10—prove to be particularly effective at improving merging performance.

| ViT-B/32 (20 tasks) | Accuracy (%) | | |
|---|---|---|---|
| **Zero-Shot** | 56.1 | | |
| **Finetuned** | 91.3 | | |
| **Zero-Shot + Distill** | 63.5 | | |
| **Merging Method** | Baseline | + Distill$_{Aux}$ | + RI (Ours) |
| **Iso-C** | 75.1 | 75.0 | 77.5 |
| **Iso-CTS** | 77.7 | 76.4 | 79.3 |
| **TSV-M** | 76.5 | 77.1 | 80.3 |

Table 4: Resolving Interference (RI) and Merging performs (green) outperforms merging followed by Multitask Distillation using auxiliary data (blue).

In contrast, datasets with low-resolution images (CIFAR100), grayscale content (FER2013), or domain-specific pathology images (PCAM) are far less effective. Interestingly, even Gaussian noise provides a noticeable boost in merging performance. These results suggest that auxiliary datasets with **high resolution and high visual diversity** are most beneficial, likely because diverse input tokens help the distillation process more accurately match the reference model's output distribution.

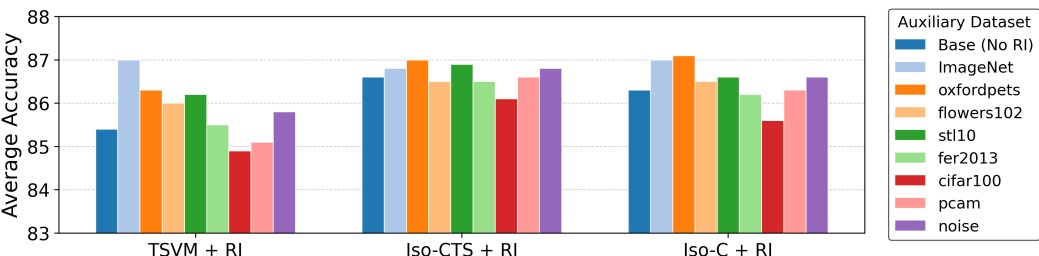

Figure 4: **What makes a good source of Auxiliary data for RI?** Datasets with colored, high-resolution images, such as ImageNet, OxfordPets, Flowers102 and STL10 perform well, while datasets with low-resolution/grey-scale/cross-domain images like CIFAR10, FER2013 and PCAM don't. Notably, passing Gaussian noise also leads to a notable improvement in merging performance.

**Are merging methods with RI sensitive to the tuning of merging hyperparameters?** It is common practice to tune merging hyperparameters—such as the scaling coefficient or pruning factor—assuming access to privileged, labeled, task-specific validation data. We compare this practice to using default merging hyperparameters, both with and without RI. As shown in Table 5, experts adapted with RI exhibit markedly lower sensitivity to hyperparameter tuning, with only a **0.4%** difference in performance on average across six different merging techniques, compared to a **1.8%** difference for unadapted models. We believe that models with reduced interference benefit from a flatter, more stable optimization landscape, allowing them to generalize well even under default hyperparameter settings and reducing the need for validation-based tuning, which may not be feasible.

| Method | Default | | Tuned on Aux Data | | Tuned on Task Data | |
|---|---|---|---|---|---|---|
| | Baseline | +RI (Ours) | Baseline | +RI (Ours) | Baseline | +RI (Ours) |
| **TA** | 56.1 | **64.7** | 60.2 | **63.4** | 61.3 | **64.5** |
| **TIES** | 58.0 | **67.7** | 61.0 | **66.1** | 63.7 | **68.9** |
| **KnOTS** | 66.7 | **69.3** | 62.3 | – | 66.6 | **70.5** |
| **Iso-C** | 75.1 | **77.5** | 67.7 | **74.6** | 75.1 | **77.4** |
| **Iso-CTS** | 77.7 | **79.3** | 65.8 | **70.9** | 77.8 | **79.3** |
| **TSVM** | 76.5 | **80.3** | 65.9 | **80.6** | 76.5 | **80.6** |
| **Mean(%)** | 68.3 | **73.1** | 63.8 | **71.1** | 70.1 | **73.5** |

Table 5: **Sensitivity to hyperparameter tuning.** Models adapted with RI are less sensitive to tuning on privileged task-specific validation data, while tuning based on cross-task interference using auxiliary data proves far less effective.

**Can the cross-task interference objective on auxiliary data be used to tune merging hyperparameters?** Not quite. As shown in Table 5, tuning merging hyperparameters with unlabeled auxiliary data using the cross-task interference KL objective from Eq. 1 generally performs worse than simply using default hyperparameters—both when adapting models with RI and when not. This degradation likely arises from overfitting to the auxiliary input distribution, which may differ substantially from the true task distribution. These findings highlight the need for new, more robust objectives for hyperparameter tuning in scenarios where task-specific validation data is unavailable. We hope the community continues to explore other proxy objectives in the absence of task-specific validation data.

## 9 CONCLUSION

We formally define the notion of *cross-task interference* as a principled diagnostic to quantify interference in model merging, capturing the representation mismatch between the merged model and its constituent experts. To minimize this, we propose *Resolving Interference* (*RI*), a lightweight adaptation strategy which adapts expert models into disjoint functional subspaces, there by reducing cross-task interference. This helps existing merging methods improve by 10% on in-domain and by 2.3% on unseen domain, demonstrating its effectiveness.

## 10 REPRODUCIBILITY STATEMENT

We encourage readers to reproduce our work, to facilitate this, we will be adding a link to our code base in the camera-ready version. Further, we share details of hyperparameters used to tune RI in Sec. 6 and share the default merging hyperparameters used in the Appendix. A.3.1. Consistent with prior

works we make use of finetuned model checkpoints for ViT-B32, and ViT-B/16, ViT-L/14 models from Wang et al. (2024c).

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

# A APPENDIX

## A.1 RI ALGORITHM

**Resolving Interference (RI )** We disentangle expert models by following the algorithmic procedure described in Alg. 1.

---
**Algorithm 1:** Resolving Interference (RI) Procedure

---
**Input:** Task-specific models $\{\theta_i = \theta_0 + \tau_i\}_{i=1}^N$, task heads $\{h_i\}_{i=1}^N$, auxiliary dataset $D_{\text{aux}}$,
      hyperparameter $\alpha$
**Output:** Disentangled task vectors $\{\tau_i^*\}_{i=1}^N$, merged model $\theta_{\text{merged}}$
**for** $i \in \{1, \ldots, N\}$ **do**
    Initialize $\tau_i^* \leftarrow \tau_i$;
    **while** *not converged* **do**
        Sample batch $x \sim D_{\text{aux}}$;
        Compute task-specific preservation loss:
            $\mathcal{L}_{\text{task}} \leftarrow \text{KL}(f(x; \theta_0 + \tau_i, h_i) \,\|\, f(x; \theta_0 + \tau_i^*, h_i))$;
        Compute interference suppression loss:
            $\mathcal{L}_{\text{interfere}} \leftarrow \frac{1}{N} \sum_{j \neq i} \text{KL}(f(x; \theta_0, h_j) \,\|\, f(x; \theta_0 + \tau_i^*, h_j))$;
        Combine losses:
            $\mathcal{L}_{\text{RI}} \leftarrow \mathcal{L}_{\text{task}} + \alpha \cdot \mathcal{L}_{\text{interfere}}$;
        Update $\tau_i^*$ via gradient descent on $\mathcal{L}_{\text{RI}}$;
    **end**
**end**
Compute $\tau_{\text{merged}} \leftarrow Merge(\tau_1^*, \ldots, \tau_N^*)$;
Set $\theta_{\text{merged}} \leftarrow \theta_0 + \tau_{\text{merged}}$;
**return** $\theta_{merged}$

---

## A.2 MERGING BASELINE DESCRIPTION

We now describe each of our merging baselines: **Weight-averaging** involves merging the task vectors by simply averaging them i.e. $\tau_{avg} = 1/N \sum_{i=1}^N \tau_i$. **Task-Arithmetic (TA)** merges task vectors by computing a linear sum: $\tau_{\text{TA}} = \sum_{i=1}^N \lambda_i \tau_i$, where $\lambda_i$ is a task-specific scaling coefficient. Since jointly tuning multiple $\lambda_i$ (one for each task) is computationally expensive, a common practice is to use a single shared scaling

| Merging Methods | 2 Tasks | 8 Tasks | 14 Tasks | 20 Tasks |
|---|---|---|---|---|
| TA | 0.42 | 0.30 | 0.22 | 0.15 |
| TIES | 1.0 | | | |
| KnOTS | 1.0 | | | |
| Iso-C | 1.9 | 1.3 | 1.o | 0.9 |
| Iso-CTS | 2.1 | 1.5 | 1.2 | 1.1 |
| TSVM | 1 | | | |

Table A1: Scaling coefficients for different numbers of tasks.

coefficient $\lambda$ for all tasks i.e. $\tau_{\text{TA}} = \lambda \sum_{i=1}^N \tau_i$. **TIES merging** minimizes conflicts between task vectors by first trimming low-magnitude weights, followed by averaging only the elements whose signs align with the elected sign. **KnOTS** jointly transforms task vectors into an aligned space using Singular Value Decomposition: $[\tau_1, \tau_2, \ldots, \tau_N] = U\Sigma[V_1, V_2, \ldots, V_N]$. In the aligned space, other merging techniques such can be TIES is applied to merge all the $V_i$'s to compute $\tau_{\text{KnOTS}} = U\Sigma V_{\text{merged}}$. **TSVM** (Gargiulo et al., 2025) formulates merging as a *Task Subspace Vector Merging* problem by first projecting each task vector $\tau_i$ onto a shared low-dimensional subspace $P$, i.e., $\tilde{\tau}_i = P^\top \tau_i$, and then performing sign–aligned averaging in this subspace to obtain the merged vector $\tau_{\text{TSVM}} = P\left(\frac{1}{N} \sum_{i=1}^N \text{sign}(\tilde{\tau}_i) \odot |\tilde{\tau}_i|\right)$. **Iso-C** (Marczak et al., 2025) performs an *isotropic combination* of task vectors by whitening their covariance, representing each task as $\hat{\tau}_i = \Sigma^{-1/2}(\tau_i - \mu)$ where $\mu = \frac{1}{N} \sum_i \tau_i$ and $\Sigma$ is the empirical covariance, and then averaging to yield $\tau_{\text{Iso-C}} = \mu + \Sigma^{1/2}\left(\frac{1}{N} \sum_{i=1}^N \hat{\tau}_i\right)$. **Iso-CTS** (Marczak et al., 2025) extends Iso-C with a cross-task scaling step by assigning each task a similarity-based coefficient $s_i$, producing the final merge $\tau_{\text{Iso-CTS}} = \mu + \Sigma^{1/2}\left(\frac{1}{\sum_i s_i} \sum_{i=1}^N s_i \hat{\tau}_i\right)$, which adaptively weights tasks according to their pairwise correlations.

### A.3 Merging Hyperparameters

### A.3.1 Default Hyperparameters

Since we operate in a data scarce setting, we make use of the recommended default merging hyperparameters for all merging baselines. Table. A1 showcases the default scaling coefficients used when merging, along with the number of tasks it is to be used for. In the case of missing recommendations for the scaling co-efficient from the original work, such as for TA in the 14 and 20 task setting and Iso_C and Iso_CTS in the 2 task setting, we extrapolate the known values logarithmically, which seems to lie close to the trend followed after tuning. Apart from scaling co-efficients, for TIES and KnOTS we set Top-K pruning factor to 20%, where onlt the top 20% of the weights are retained across each task-vector. For Iso-CTS, we set common-space-fraction to 0.8.

### A.3.2 Tuning on Task Data or Auxiliary data

For the case when we do tune the merging hyperparameters, we do the sweep across the following range and stop when the average validation accuracy or the RI Loss drops:

TA: Scaling co-efficient: 30 intermediate steps $\in [0, 1]$
TIES and KnOTS: Scaling co-efficient 30 intermediate steps $\in [0, 3]$, 10 intermediate steps $\in [10, 100]$
Iso_C: Scaling co-efficient 30 intermediate steps $\in [0, 3]$
Iso_CTS: Scaling co-efficient 30 intermediate steps $\in [0, 3]$, Common Space Fraction: 6 intermediate steps $\in [0.5, 1.0]$

### A.4 8/14/20 vision tasks

The 8/14/20 task vision benchmark includes the following: 1. Cars (Krause et al., 2013), 2. DTD (Cimpoi et al., 2014), 3. EuroSAT (Helber et al., 2019), 4. GTSRB (Stallkamp et al., 2011), 5. MNIST (LeCun, 1998), 6. RESISC45 (Cheng et al., 2017), 7. SUN397 (Xiao et al., 2016), 8. SVHN (Netzer et al., 2011), 9. CIFAR100 (Krizhevsky et al., 2009), 10. STL10 (Coates et al., 2011), 11. Flowers102 (Nilsback & Zisserman, 2008), 12. OxfordIIITPet (Parkhi et al., 2012), 13. PCAM (Veeling et al., 2018), 14. FER2013 (Goodfellow et al., 2013), 15. EMNIST (Cohen et al., 2017), 16. CIFAR10 (Krizhevsky et al., 2009), 17. Food101 (Bossard et al., 2014), 18. FashionMNIST (Xiao et al., 2017), 19. RenderedSST2(Socher et al., 2013), 20. KMNIST (Ba et al., 2016). Where tasks 1-8, 1-14 and 1-20 constitute the 8/14/20 task evaluations, respectively.

