# OpenReview forum: "Resolving Interference (RI): Disentangling Models for Improved Model Merging"
_ICLR.cc/2026/Conference — Submitted to ICLR 2026_

### Official Review · Reviewer_SjwS · 2025-10-27

**Soundness:** 4
**Presentation:** 4
**Contribution:** 3
**Rating:** 4
**Confidence:** 5

**Summary:**

This paper introduces Resolving Interference (RI), a lightweight framework to reduce cross-task interference in model merging, defined as merged models’ representation drift from constituent task-specialized models. RI disentangles expert models into functionally orthogonal subspaces using only unlabeled auxiliary data (no task data needed), enhancing existing merging methods.

**Strengths:**

1.The paper is well organized and well written.

2.The authors present a well-motivated approach with a simple, easy-to-follow framework.

3.It conducts numerous experiments, validates the experimental results on models of various series and sizes, and covers a wide range of evaluation tasks

**Weaknesses:**

1. **Although the paper claims to address the scenario where no model-related data are available for improving model fusion, its contribution remains unclear.**
   This is because the proposed method still requires training with additional data, and the source of this auxiliary data is restricted. The comparison settings in the paper do not convincingly isolate the true contribution of the proposed approach, as the method still depends on external data. Moreover, Figure 4 shows that not all auxiliary datasets lead to performance gains, which raises questions about the selection of these datasets (e.g., whether they are similar to the task-specific data). It is recommended that the authors include an analysis of the correlation between the auxiliary and task-specific data, and add experiments in Tables 1 and 2 using *noisy* or *irrelevant* auxiliary data (as in the purple items of Figure 4) to verify that the proposed method can indeed improve model fusion performance **without relying on task-related data**.


2. **The explanation for why weight averaging fails in the in-domain setting is not convincing.**
   The authors attribute this to the small scaling coefficient. However, in the 20-task setting, the coefficient for TA is also small, yet RI still performs effectively. Therefore, this explanation does not sufficiently account for the observed phenomenon. The authors are encouraged to provide a deeper analysis or additional experiments to investigate this behavior, rather than offering a potentially incomplete conclusion.

3. **The computational and memory costs introduced by training the proposed method should be more rigorously discussed.**
   Since the method requires an additional training phase, its practical feasibility needs further evaluation. The paper should include detailed comparisons of GPU memory usage, training time, and computational cost, as well as an analysis of scalability when the number of tasks increases (as the method likely requires more training steps in such cases).

**Questions:**

1. Would the proposed method fail or degrade in performance when applied to text classification or text generation tasks, where the output representations may be more complex than in the current settings?

2. Although the method does not require task-specific data, the authors could still use such data as auxiliary input to further validate its effectiveness and compare it with other training-based approaches. This would strengthen the paper’s contribution. I am also curious about the method’s performance when in-domain data are used as auxiliary data.

---

> ### Author Response · Authors · 2025-12-04
> **Author Response to Reviewer 4 (SjwS) (1/3)**
>
> We sincerely thank Reviewer 4 (SjwS) for the thoughtful and constructive feedback. We appreciate the recognition that the paper is well organized and clearly written, and that our method is well motivated and presented through a simple, easy-to-follow framework. We are also grateful for the acknowledgment of the technical soundness of our approach and the strong experimental validation, including the breadth of experiments across models of different sizes and series and the wide coverage of evaluation tasks. Below we address each of reviewers’ concerns.
>
> **R4W2: The explanation for why weight averaging fails in the in-domain setting is not convincing. Further analysis or evidence is needed.**
>
> **AR4W2**: In order to better understand why RI underperforms, we analyse the influence of the scaling coefficient on the merging performance. We test this across the ViT-B/32, 20-task setting.
>
> **Table A: Accuracy (in %) of Averaging+RI with varying scaling co-efficient on ViT-B/32 20 task setting.**
>
> Accuracy with Averaging = 61.0%
>
> | Scaling Coefficient | 0.0 | 0.025 | 0.05 | 0.075 | 0.1 | 0.125 | 0.15 | 0.175 | 0.2 | 0.225 | 0.25 | 0.275 | 0.3 |
> |--------------------|-----|-------|------|-------|-----|--------|------|--------|-----|--------|------|--------|-----|
> | Averaging+RI   | 56.1 | 59.3  | 61.4 | 62.9  | 64.0 | 64.6   | 64.7 | 64.3   | 63.4 | 62.1   | 60.4 | 58.6   | 56.6 |
>
>
> As seen in Table A, the gains with RI aren’t visible at a low scaling coefficient of 0.05 used by averaging. In contrast, Task Arithmetic—which is algebraically equivalent to Averaging but uses a relatively larger coefficient of 0.15, leads to higher gains. Averaging with extremely low coefficients underutilises RI’s corrective signal rather than contradicting its utility.
> Moreover, while Averaging is included for completeness, it is widely regarded as the weakest baseline among modern merging techniques. Our empirical results show that RI consistently improves all strong baselines (TA, TIES, KnOTS-TIES, Iso-C, Iso-CTS, TSVM, WUDI), which are the primary evaluation focus of this work.
>
> **AR4W3: The computational and memory costs introduced by training the proposed method should be more rigorously discussed.**
>
> We thank the reviewer for raising this point. While RI introduces an additional training phase, it is explicitly designed to be lightweight and parallelizable. Each expert is adapted independently using unlabeled auxiliary data, with no joint optimization across experts, making RI embarrassingly parallel and easy to scale across multiple GPUs.
>
> Compute and Memory Profile
> Using an NVIDIA A40 GPU (48GB VRAM), we report the following runtime and memory usage:
>
> **Table B: Computational Profile of RI**
>
> | Tasks | Training Steps | Time per Expert | Peak GPU Memory |
> |-------|----------------|-----------------|-----------------|
> | 8     | 2500           | 7m 07s          | 4.8 GB |
> | 14    | 2500           | 8m 33s          | 4.8 GB |
> | 20    | 2500           | 8m 50s          | 4.8 GB |
>
> Using standard efficiency techniques (data prefetching and torch.compile), per-expert runtime was reduced from approximately 15 minutes (as originally stated in L259) to about 7 minutes for ViT-B/32 in the 8-task setting. Notably, even when scaling to 20 tasks, runtime increases by less than 2 minutes, while memory usage remains constant. For comparison, training-data–based merging methods which perform joint optimization are significantly more expensive: Adamerging [1] requires 127 minutes and 17.1 GB of GPU memory in the 8-task setup reported in [3].
>
> **Scalability Analysis**
> Adapting a single expert with RI involves O(S) backbone forward passes and O(NS) head forward passes, where N is the number of tasks and S the number of training steps. In CLIP models, task heads are single linear layers, so this additional cost is minimal. As shown in Table B, runtime grows sub-linearly with the number of tasks.
>
> Regarding training steps, RI does not require more optimization iterations as tasks increase. As shown in Table 1, we use the same number of training steps for 8, 14, and 20 tasks, yet observe increased gains at higher task counts, demonstrating scalability without increased compute.
> We will incorporate this computational analysis into the final version of the paper.

---

> ### Author Response · Authors · 2025-12-04
> **Author Response to Reviewer 4 (SjwS) (2/3)**
>
> **R4Q1: Would the proposed method fail or degrade in performance when applied to text classification or text generation tasks, where the output representations may be more complex than in the current settings?**
>
> **AR4Q1:**  The RI objective in Eq. 4 is defined over the output representations of a model (e.g., logits or predicted outputs) and makes no modality- or architecture-specific assumptions. RI is therefore directly applicable to transformer-based LLMs and other NLP architectures. For example, in the case of LLMs, the head could correspond to the LM head / unembedding matrix that maps hidden states to token logits, where the RI loss can be applied on the logits corresponding to the predicted tokens.
>
> **AR4Q2 — Resolving Interference with In-Domain Task Data**
> We thank the reviewer for this valuable suggestion. Evaluating RI with access to in-domain task data provides important insight into the upper bound of its effectiveness when distribution shift is removed. In Table C, we report the results of applying RI directly on task data in the 8-task ViT-B/32 vision setting. Since task data is available in this setting, we additionally tune all merging hyperparameters instead of using default values.
>
> **Table C: RI on Task-Data (8-task, ViT-B/32)**
>
> | Merging Method | Baseline | RI on Aux | RI on Task |
> |----------------|----------|-----------|------------|
> | Average | 66.1 | 65.7 | 64.4 |
> | TA | 69.2 | 76.5 | 79.9 |
> | TIES | 73.7 | 79.3 | 82.7 |
> | KnOTS | 77.2 | 78.4 | 83.6 |
> | Iso-C | 86.3 | 87.0 | 87.1 |
> | Iso-CTS | 86.6 | 86.8 | 87.6 |
> | TSVM | 85.4 | 87.0 | 88.7 |
>
> Using task data yields an additional performance gain of **+3.4%** for TA and TIES, **+5.2%** for KnOTS, and **+1.7%** for TSVM, demonstrating that RI can further benefit when auxiliary and target distributions are aligned. These results highlight the performance gap introduced by the distribution shift.
> Importantly, despite having no access to task data, RI with auxiliary data still captures the majority of the attainable gains, achieving strong improvements across all major merging techniques. This validates RI’s practical value in realistic settings where task data may be unavailable, while showing that additional gains are possible when distribution alignment is favourable.
>
> We additionally compare RI with task-data–based merging approaches using results reported in the original works on Adamerging and Model Surgery. In this setting, RI achieves 79.9%, which is slightly lower than Adamerging (80.1%) and Model Surgery (80.9%), as shown in Table D. More broadly, we believe this setting is less central to the merging problem, as the availability of task data allows alternative paradigms—such as joint multi-task training or distillation—which are known to achieve even stronger performance, as reported in [2].
>
> **Table D: Comparison of RI with task-data–based training approaches**
>
> | Method | Accuracy |
> |--------|----------|
> | TA | 66.3 |
> | TA + RI | 79.9 |
> | Adamerging | 80.1 |
> | Model Surgery | 80.9 |
>
> We will include this comparison in the final version to clarify RI’s behavior across both in-domain and out-of-domain regimes.

---

> ### Author Response · Authors · 2025-12-04
> **Author Response to Reviewer 4 (SjwS) (3/3)**
>
> **R4W1: Although the paper claims to address the scenario where no model-related data are available for improving model fusion, the proposed method still requires training with additional data, making its contribution remain unclear.**
>
> **ARW1.a:**
> We would like to clarify that our contribution is not that RI operates without any data, but rather that it does not require *task-specific data* to adapt the expert models being merged. This is precisely the regime where prior gradient-based methods [1,2] are largely inapplicable, as they assume access to task data.
>
> RI instead relies on arbitrary, unlabeled auxiliary data that is not tied to any task or expert—this is a qualitatively weaker and more realistic assumption. Our experiments explicitly demonstrate that RI remains effective even when the auxiliary inputs are semantically unrelated to the target tasks, including when using Gaussian noise as input (Figure 4). This indicates that RI is not learning task semantics from auxiliary data, but instead uses the auxiliary inputs purely as a probe distribution to enforce representation-level functional orthogonality among experts.
>
> Thus, our key contributions are as follows:
>
> - We propose Resolving Interference (RI), a lightweight framework that disentangles expert models into functionally orthogonal subspaces using auxiliary inputs with high diversity.
> - We show that RI improves merging performance by up to 9.7% across diverse benchmarks and model scales, and enhances out-of-distribution robustness by up to 2.3%.
> - We conduct extensive ablations over data sources and optimization strategies, offering practical guidance for applying RI effectively.
>
> ---
>
> **RW1.b: Figure 4 shows that not all auxiliary datasets lead to performance gains, raising questions about how these datasets are chosen and whether they are related to task-specific data. An analysis of correlation between auxiliary and task data distributions is recommended.**
>
> **ARW1.b:**  To quantify the relationship between auxiliary and target datasets, we compute Maximum Mean Discrepancy (MMD²)  between their corresponding ViT-B/32 OpenCLIP embeddings. A lower MMD² score indicates greater similarity, with MMD² = 0 implying identical distributions. The full results are shown in Table A, along with TSVM merging accuracy as a reference metric.
>
> The Pearson correlation between the average MMD² score across target tasks and TSVM merging accuracy is **−0.406**, indicating weak correlation between dataset similarity and merging performance. While high-resolution and diversity-rich datasets tend to perform better in practice, the results suggest that auxiliary–task similarity alone does not strongly predict merging effectiveness. Understanding how auxiliary data properties influence optimization remains an interesting direction for future work.
>
> **Table E: MMD² score across different pairs of Auxiliary and Target data sources (lower implies higher correlation)**
>
> | Auxiliary Data → \ Target Data | imagenet | cifar100 | stl10 | flowers102 | oxfordpets | pcam | fer2013 |
> |-------------------------------|----------|----------|--------|------------|------------|------|---------|
> | stanford_cars | 0.420559 | 0.802594 | 0.486463 | 0.902734 | 0.753621 | 1.402615 | 0.977105 |
> | dtd | 0.220334 | 0.436505 | 0.443429 | 0.554313 | 0.636759 | 1.059627 | 0.711119 |
> | eurosat | 0.65531 | 0.554339 | 0.67467 | 1.166788 | 1.075644 | 1.503099 | 0.924496 |
> | gtsrb | 0.659394 | 0.526733 | 0.726759 | 1.209394 | 1.046505 | 1.581054 | 0.841615 |
> | mnist | 1.097324 | 0.864856 | 1.167628 | 1.571174 | 1.488189 | 1.930499 | 1.052053 |
> | resisc45 | 0.444175 | 0.651189 | 0.531873 | 0.939804 | 0.884912 | 1.294175 | 0.914043 |
> | sun397 | 0.110982 | 0.469979 | 0.266017 | 0.591539 | 0.531041 | 1.095848 | 0.679938 |
> | svhn | 0.879087 | 0.605029 | 0.966995 | 1.386325 | 1.247799 | 1.730513 | 0.928676 |
> | **Average** | 0.56027 | 0.61315 | 0.65723 | 1.04026 | 0.95806 | 1.44968 | 0.87863 |
> | **TSVM Merging Accuracy** | 87.0 | 84.894 | 86.249 | 86.033 | 86.337 | 85.082 | 85.461 |
> ---
>
> **RW1.c: Add experiments in Tables 1 and 2 using noisy or irrelevant auxiliary data (as in Figure 4) to verify that the method does not rely on task-related data.**
>
> **ARW1.c:**  Our primary focus in Tables 1 and 2 is the task-data–free setting using real auxiliary data. We do not claim that Gaussian noise is optimal; rather, it serves as a stress test demonstrating RI’s robustness to auxiliary distribution choice. The results in Figure 4 already show that RI remains effective even with noise, supporting our central claim that RI does not rely on task-related auxiliary information.
>
> References:
> [1] Yang, Enneng, et al. "Adamerging" arXiv preprint arXiv:2310.02575 (2023).
> [2] Yang, Enneng, et al. "Representation surgery" arXiv preprint arXiv:2402.02705 (2024).
> [3] Cheng, Runxi, et al. "Whoever started the interference should end it" arXiv preprint arXiv:2503.08099 (2025).

---

### Official Review · Reviewer_1att · 2025-10-31

**Soundness:** 2
**Presentation:** 3
**Contribution:** 2
**Rating:** 2
**Confidence:** 4

**Summary:**

This paper introduces the concept of cross-task interference in model merging: the merged model underperforms its constituent experts.
It proposes a training-based merge method that defines the interference as the KL divergence between the merged and expert models and trains on unlabeled auxiliary data to reduce this cross-task interference. The authors demonstrate that RI improves standard merging across 8/14/20-task benchmarks on ViT-B/32, B/16, and L/14.

**Strengths:**

- This paper tackles merging interference by matching the expert output with the merged model output and proposes a new Resolving Interference (RI) method.
- The paper conducts up to 20 datasets experiments and presents improved results on CLIP across various merging methods.
- It includes a detailed analysis of the use of an auxiliary dataset, which requires no access to labels.

**Weaknesses:**

- The weaknesses of RI loss:
  - It only applies to classification tasks and ignores generative tasks and LLMs (which are dominant): all experiments use CLIP-style ViT encoders and vision classification heads.
  - The algorithm explicitly requires the set of heads $\{h_i\}_{i=1}^N$. This assumption does not hold for many generative settings (e.g., segmentation, detection, diffusion, LLMs).
  - It cannot handle different task types, which are naturally supported by other merging techniques (e.g., summarization, math, and code tasks can be merged via Task Arithmetic).
  - It introduces an unnecessary pipeline procedure: it adds a gradient-based adaptation stage for **each expert** (RI), instead of a single gradient-free merge. By default, it uses 2,500 steps (320k samples) on auxiliary data **per expert**, which incurs high training cost, whereas other merging techniques (e.g., Task Arithmetic, TIES) are training-free.
  - Assuming N tasks, RI requires $O(N)$ forward passes per step per expert; across all experts and S steps, this becomes $O(N^2S)$  head forwards plus $O(NS)$ backbone forwards.
    - When the number of tasks is very large, the loss incurs high computational complexity.
    - As model size scales, each forward pass becomes more expensive, making RI even costlier.
    - In Figure 3, the paper shows an “elbow” around 2,500 steps. With a maximum task number of N = 20 and S = 2,500 RI steps, this requires >1M head forwards and >50k backbone forwards in total.
- The weaknesses of auxiliary data:
  - The use of arbitrary auxiliary data is justified empirically rather than theoretically. The authors show that even Gaussian noise yields a noticeable performance boost. Why does directly using noise improve model performance? A noisy dataset may drive expert models to produce meaningless outputs, which should not benefit model training.
  - It implicitly assumes that the auxiliary data comes from the same task type. It remains unclear whether auxiliary data can still be used when merging different task types.
- The weaknesses of the experiments:
  - Averaging + RI (Ours) consistently underperforms plain Averaging on both ViT-B/16 and ViT-L/14 across 8, 14, and 20 tasks.
  - The evaluation does not cover most SOTA baselines, such as Surgery, TwinMerging, CatMerging, and LoTMerging.
- Typo: Line 281: “78.7KnOTS”

**Questions:**

See Weakness

---

> ### Author Response · Authors · 2025-12-04
> **Author Response to Reviewer 3 (1att) (1/3)**
>
> We sincerely thank the Reviewer 3 (1att) for the detailed and thoughtful assessment of our work. We appreciate the recognition of (i) the formulation of cross-task interference, (ii) the RI framework, and (iii) our extensive experimental analysis across up to 20 tasks on multiple CLIP backbones.
>
> Below, we address each of the reviewer’s concerns in detail and provide additional experiments and clarifications to further strengthen the paper. We also believe that some points in the review may stem from misunderstandings, which we clarify below.
>
> ---
>
> **R3S1: “This paper tackles merging interference by matching the expert output with the merged model output”**
>
> **ARS1:** We would like to clarify a potential misunderstanding: RI does **not** match the merged model's representations with expert models. Rather, Resolving Interference (RI) is an adaptation technique applied before the models are even merged and is agnostic to the choice of merging technique.
>
> **We briefly summarise how RI works below:**
> RI enforces the RI loss in Eq. (4), which has two objectives (shown in red and blue in the paper).
>
> - The **blue objective** ensures that the task vectors corresponding to each expert model have no influence on the output representation across the heads of other tasks.
> - The **red objective** ensures that the adapted expert preserves its own expertise by matching outputs produced through its own head with the unadapted expert.
>
> Once experts are adapted with RI, they may be merged using any existing merging method. Unlike other adaptation strategies like Adamerging [1] and Model Surgery [2], RI does not require task data and only uses unlabeled auxiliary data, making it suitable in settings where task data may be unavailable.
>
> ---
>
> **R3W1: RI Loss**
>
> **R3W1.a: “RI only applies to classification tasks; cannot handle generative tasks or LLMs.”**
>
> **AR3W1:** We would like to clarify that while RI is evaluated in classification settings, its formulation does not assume classification. The RI objective in Eq. 4 is defined over the output representations of a model (e.g., logits or predicted outputs) and therefore extends naturally to generative models as well.
>
> For example, in the case of LLMs, we can use the logits corresponding to the predicted token or utilize predicted pixels in the case of diffusion models.
>
> While this work focuses on formulating a lightweight, task-data-free adaptation strategy to resolve interference and tests it on popular CLIP-based merging benchmarks, the approach can be naturally extended to both classification and generative models.
>
> ---
>
> **R3W1.b: “The algorithm requires a set of task heads; this does not hold in generative settings (e.g., segmentation, detection, diffusion, LLMs).”**
>
> **AR3W1.b:**
> We define a task head in a broad and architectural sense as the final task-specific interface through which a model's intermediate representations are mapped to observable outputs. This need not be restricted to a classification head.
>
> For example, in LLMs, the head could correspond to the LM head / unembedding matrix that maps hidden states to token logits. In diffusion models, the role of the “head” is naturally played by the decoder, e.g., the VAE decoder in latent diffusion, which maps latent representations to the image domain. Similarly, for structured prediction models such as detection or segmentation, the head corresponds to the task-specific prediction modules (e.g., object query decoders or decoders that output dense predictions). In general, our framework assumes only the presence of a distinct task-output interface.
>
> ---
>
> **R3W1.c: Cannot handle different task types (e.g., summarization, math, code) supported by other merging techniques such as Task-Arithmetic.**
>
> **AR3W1.c:** As explained in R3W1.b, RI naturally extends to LLMs by treating the unembedding layer as the task head and computing the RI loss on token-level output distributions. Since RI is defined purely in terms of output distributions, it is inherently task-agnostic and does not depend on whether the model’s expertise is in summarization, math, or code.
>
> Importantly, RI itself is not a merging algorithm, but an interference-reduction framework that can be paired with merging techniques such as Task-Arithmetic. We will add this clarification regarding handling different task-types to the main paper.

---

> ### Author Response · Authors · 2025-12-04
> **Author Response to Reviewer 3 (1att) (2/3)**
>
> **R3W1.d: “RI introduces unnecessary overhead; it is costly for large N and large models.”**
>
> **AR3W1.d:**
> We respectfully disagree that RI is unnecessary or impractical. The computational cost is justified by substantial performance gains over existing merging techniques, as detailed below.
>
> RI yields up to **+9.7%** improvement over TA and TIES, up to **+11.1%** over WUDI merging (as seen in AR1W2), and up to **+3.9%** over other SOTA (TSVM) gradient-free merging methods across the 20-task vision benchmark, as seen in Table 1. It also helps improve the merged model's generalisation performance by up to **+2.3%** as seen in Table 2.
>
> A key empirical pattern across all architectures is that merging performance degrades sharply as the number of tasks increases. RI consistently mitigates this degradation by reducing cross-task interference as seen in Figure 3 (middle) and becomes increasingly beneficial as the task count grows.
>
> **Reasonably Lightweight**
>
> Additionally, by using common efficiency tricks—prefetching data and compiling the PyTorch model—we significantly reduced per-expert RI adaptation time from ~15 minutes (as stated in L259) to just ~7 minutes in the 8-task ViT-B/32 setting, while scaling up to just under 9 minutes per expert when scaling up to 20 tasks, with stable GPU memory usage. Here’s a complete compute cost profile of RI.
>
> **Table B: Computational Profile of RI**
>
> | Tasks | Training Steps | Time per Expert | Peak GPU Memory |
> |-------|----------------|-----------------|-----------------|
> | 8     | 2500           | 7m 07s          | 4.8 GB          |
> | 14    | 2500           | 8m 33s          | 4.8 GB          |
> | 20    | 2500           | 8m 50s          | 4.8 GB          |
>
> For comparison, popular training-data-based adaptation methods such as **Adamerging [1] jointly adapt task-vectors requiring 127 minutes and 17.1 GB memory in the 8-task vision setting** as tested in [3]. Therefore, the added training cost of RI is justified based on the performance improvement and relatively lower computational requirement.
>
> ---
>
> **R3W1.d: “Assuming N tasks, RI requires O(N) forward passes per step per expert; across all experts and S steps, this becomes O(N²S) head forwards plus O(NS) backbone forwards.”**
>
> We would like to clarify that when adapting a single expert, each step only incurs a fixed 3 backbone forward passes and N head forward passes. Thus, for each expert, it takes O(S) backbone forwards and O(NS) head forwards. This is further explained below:
> - **Red Objective:** Requires a backbone forward and task-head forward pass through the unmodified expert and the adapted expert model.
> - **Blue Objective:** Reuses the embeddings of the adapted expert from above and incurs a third backbone forward pass through the pretrained model, followed by head forward passes through the remaining N−1 heads. Note, it is possible to reuse the backbone embeddings when adapting all expert models.
>
> In CLIP models, heads are single linear layers, incurring minimal cost. Table B shows that per-expert adaptation increases from 7m07s to only 8m50s when increasing tasks from 8 to 20. Since decoder layers are lighter than backbones, overhead scales sub-linearly. We will add this discussion to the main paper.
>
> ---
>
> **R3W2: Auxiliary Data**
>
> **R3W2.a: “The use of arbitrary auxiliary data is justified empirically rather than theoretically.”**
>
> It is true that our method is currently justified empirically rather than theoretically. In Figure 3, we motivated our work by demonstrating that minimizing the RI loss on auxiliary data leads to a proportional reduction in cross-task interference measured on task data, establishing a direct empirical link between optimization on auxiliary data and the desired effect on target tasks.
>
> Prior work has reported similar observations, showing that alternative data sources—such as unrelated real data or even synthetic data (e.g., Gaussian noise)—can be effective for knowledge transfer and distillation [4]. While developing a formal theoretical understanding of when and why objectives optimized on auxiliary data transfer to task data is an important direction for future work, a detailed theoretical treatment is beyond the scope of this paper.
>
>
> ---
>
> **R3W2.b: “Why does Gaussian noise help?”**
>
> We do not claim Gaussian noise is optimal; rather, it serves as a stress-test demonstrating RI's robustness to auxiliary data distribution. Prior work [4] shows that even synthetic or unrelated data (Eg, Gaussian Noise, OpenGL, Leaves) can support knowledge distillation. Since RI does not rely on semantic content or labels, but only on inducing diverse activations we hypothesize that even noise is sufficient to distil functional orthogonality across task-spaces. The fact that structured, diverse datasets (ImageNet, OxfordPets) significantly outperform noise (Figure 4) confirms that semantic richness improves results, though is not strictly required.
>
> ---

---

> ### Author Response · Authors · 2025-12-04
> **Author Response to Reviewer 3 (1att) (3/3)**
>
> **R3W2.c: “It implicitly assumes auxiliary data comes from the same task type.”**
>
> **AR3W2.c:** RI does not make any assumption about which task type the auxiliary data (Eg, summarization, math, code) belongs to. The performance improvement we observe when using Gaussian noise as the source of auxiliary data further confirms this.
> While we do not explicitly test across settings with different task types, our formulation of RI loss, as stated in Eq.4, naturally extends to such models by using the task-specific head/decoder to obtain the logits/output probabilities for distillation objectives in RI Loss.
>
>
> ---
>
> **R3W3.a: “Averaging + RI underperforms Averaging on ViT-B/16 and ViT-L/14.”**
>
> **AR3W3.a:** We discuss this in L312-315. It is due to the fact that the scaling coefficient associated with averaging is too low. We provide further evidence of this by showcasing how the performance with RI changes by varying the scaling coefficient with averaging+RI. We test this across the ViT-B/32, 20-task setting. As seen in Table C, the gains with RI aren’t visible with a low scaling coefficient of 0.05 used by averaging, whereas Task-Arithmetic, which works the same as averaging, but uses a higher scaling coefficient of 0.15, leads to higher gains.
>
> **Table C: Accuracy (in %) of Averaging+RI with varying scaling co-efficient on ViT-B/32 20 task setting**
>
> | Scaling Coefficient | 0.0 | 0.025 | 0.05 | 0.075 | 0.1 | 0.125 | 0.15 | 0.175 | 0.2 | 0.225 | 0.25 | 0.275 | 0.3 |
> |--------------------|-----|-------|------|-------|-----|--------|------|--------|-----|--------|------|--------|-----|
> | Averaging+RI   | 56.1 | 59.3  | 61.4 | 62.9  | 64.0 | 64.6   | 64.7 | 64.3   | 63.4 | 62.1   | 60.4 | 58.6   | 56.6 |
>
> Averaging model weights is the simplest and generally the least performant merging technique, where RI does not showcase consistent performance improvement. Note, this is not the case when testing for generalization as seen in Table 2, where Averaging+RI improves generalization performance by 2.0%. We will add this discussion to the appendix section of the main paper.
>
> ---
>
> **R3W3.b:“The evaluation does not cover most SOTA baselines, such as Surgery, TwinMerging, CatMerging, and LoTMerging.”**
>
> **AR3W3.b:** Our setting assumes no task data:
>
> AR3W3.b: We discuss the selection of merging baselines in the related works section in L107-112 and L241-244. Our focus in this work is on the setting where no-task data is available. We discuss below the decision behind why these methods were excluded as baselines.
>
> **Model Surgery:** requires training task-specific modules, which not only require additional capacity and increase latency but also access to task-data, rendering it out of scope for our task-data free setting.
>
> **Twin-merging:** Involves an MoE like dynamic routing/composition of task-specific modular components and does not produce a single shared model. Additionally it require training a router on task-data and hence regarded out of scope.
>
> **CatMerging and LoTMerging:** While these methods are certainly in scope the past year has seen a large number of merging methods being designed. Unable to test our method across all of them we picked a mix of prominent methods like Averaging, TA, TIES, KnOTS and SOTA methods like TSVM, Iso-C and Iso-CTS. Moreover our selected SOTA merging baselines TSVM, Iso-C and Iso-CTS and newly added WUDI merging all showcase higher performance than LOT and CAT Merging as seen in the popular ViT-B/32 8 task benchmark. Based on this evidence, we decided to test our method across the merging method with the highest performance.
>
> ---
>
> **R3W4: Typo in L281**
>
> **AR3W4:** We thank the reviewer for pointing out this typo. We will update it in the main paper.
>
> ---
>
> **References**
>
> [1] Yang, Enneng, et al. "Adamerging: Adaptive model merging for multi-task learning." arXiv preprint arXiv:2310.02575 (2023).
>
> [2] Yang, Enneng, et al. "Representation surgery for multi-task model merging." arXiv preprint arXiv:2402.02705 (2024).
>
> [3] Cheng, Runxi, et al. "Whoever started the interference should end it: Guiding data-free model merging via task vectors." arXiv preprint arXiv:2503.08099 (2025).
>
> [4] Frank, Logan, and Jim Davis. "What Makes a Good Dataset for Knowledge Distillation?." Proceedings of the Computer Vision and Pattern Recognition Conference. 2025.

---

### Official Review · Reviewer_FeW7 · 2025-10-31

**Soundness:** 2
**Presentation:** 3
**Contribution:** 3
**Rating:** 4
**Confidence:** 4

**Summary:**

The paper proposes resolving interference (RI) to reduce cross-task interference for model merging, with the potential to utilize auxiliary input. The proposed method outperforms reported baselines when merging ViT with 8, 14, and 20 tasks. Overall, the writing and organization of the paper are good.

**Strengths:**

1) The paper proposes a distillation method to disentangle the output of a task-specific model through the definition of cross-task interference.
2) The proposed method is examined in vision tasks following the model merging recent setup with 8, 14, and 20 tasks, and it showed that it can improve the existing model merging method.
3) The experiment and ablation are well thought out and analyzed to investigate the cross-task interference and the proposed RI.

**Weaknesses:**

1) In the abstract, the 10% mentioned is confusing. The improvement made by this paper against SOTA is roughly within 2%. This is misleading.
2) Missing analysis of the data-less model merging SOTA "WUDI" merging. By how much can RI improve WUDI?
3) The proposed method is not evaluated on NLP tasks, which are commonly studied in almost all recent model merging methods. Is RI applicable to NLP tasks?
4) The computational requirements should be analyzed, and jointly optimizing the task vectors simultaneously could be very costly.

**Questions:**

Just curious. Why $\tau_i^*$ need to be optimized separately for each task? Can't we optimize a single $\tau ^ *$ for all tasks?

---

> ### Author Response · Authors · 2025-11-25
> **Author Response to Reviewer R2 (FeW7) (1/2)**
>
> We sincerely thank Reviewer R2 (FeW7) for the thoughtful and constructive assessment. We are glad the reviewer found our investigation into reducing cross-task interference, along with the accompanying experiments and ablations, to be well designed and thorough. We also appreciate the positive remarks regarding the writing quality and organization of our work.
>
> ---
>
> ### **R2W1:** *“The abstract says 10% improvement, but gains over SOTA are around 2%.”*
>
> **AR2W1:**
> The goal of Resolving Interference (RI) is to improve the merging performance of any existing merging method. The “up to 10%” improvement in the abstract refers specifically to prominent merging baselines such as Task-Arithmetic (TA) and TIES, both of which suffer significantly from cross-task interference (as also evident in Figure 1). While these methods are not state-of-the-art, they remain widely used and offer a useful lens for studying interference.
>
> From a practical standpoint, we agree that improvements over SOTA merging methods are more relevant. We will therefore update the abstract to highlight improvements of up to **3.9% on TSVM (Table 1)**, which better reflects our strongest SOTA results while preserving the broader investigative context of RI.
>
> ---
>
> ### **R2W2:** *“Missing analysis of model merging SOTA WUDI. How much does RI improve WUDI?”*
>
> **AR2W2:**
> We appreciate the suggestion. We evaluated **WUDI + RI** across all **8/14/20-task** setups for **ViT-B/32**, **ViT-B/16**, and **ViT-L/14**. The results are shown below.
>
> #### **Table A: Evaluating WUDI merging with RI on the vision 8/14/20 task benchmark**
>
> | Merging Method |          |   ViT-B/32     |        | |          |  ViT-B/16      |        | |          |  ViT-L/14      |        |
> |----------------|----------------------|--------|--------|-|----------------------|--------|--------|-|----------------------|--------|--------|
> |                               | 8 Tasks  | 14 Tasks | 20 Tasks | |   8 Tasks | 14 Tasks | 20 Tasks | | 8 Tasks            | 14 Tasks | 20 Tasks |
> | wudi                      | 82.0       |      73.5    |      58.4   | |    85.2     | 77.2   | 63.8   | | 91.0               | 84.9   | 76.2   |
> | wudi + RI (Ours)    | **84.1**  |   **77.6** |   **68.0**  | |   **86.9** | **81.6** | **74.9** | | **91.6**           | **86.3** | **82.4** |
>
>
> RI improves WUDI by **+2.1%, +4.1%, and +9.7%** on ViT-B/32 (8/14/20 tasks) and shows similarly strong improvements on larger architectures. These results confirm that RI consistently enhances even the strongest dataless merging techniques, including those explicitly designed to reduce interference. We will integrate these results into Table 1 in the main paper.
>
> Note: The WUDI baseline accuracy (82.0%) differs from the 85.2% reported in the original WUDI paper because the two works rely on different model checkpoints. WUDI uses checkpoints from [2], whereas our experiments use the 8/14/20-task checkpoint suite released by [3], which provides ViT-based OpenCLIP models across all task configurations.
>
> ---
>
> ### **R2W3:** *“No NLP experiments. Is RI applicable to NLP tasks?”*
>
> **AR2W3:**
> Our method requires only **logits from task heads** and makes no modality- or architecture-specific assumptions. RI is therefore directly applicable to **Transformer-based LLMs** and other NLP architectures.
>
> We appreciate this suggestion and agree that such an evaluation would strengthen the work. Unfortunately, we were constrained by available computing resources for this submission. We note that several well-regarded papers at this conference, including Adamerging [1], have similarly focused their empirical analysis on vision benchmarks.
>
> ---
>
> ### **R2W4:** *“Computational requirements are unclear; jointly optimizing task vectors may be expensive.”*
>
> **AR2W4:**
> RI is lightweight, parallelizable, and requires only **minutes per expert**. Importantly, **no joint optimization** is performed. Each expert is adapted independently, making RI embarrassingly parallel across experts.
>
> In addition to the compute details mentioned in L257–259, we provide a more detailed profile below (using NVIDIA A40, 48 GB VRAM):
>
> #### **Table B: Computational Profile of RI**
>
> | Tasks | Training Steps | Time per Expert | Peak GPU Memory |
> |-------|----------------|-----------------|-----------------|
> | 8     | 2500   | 7m 07s          | 4.8 GB          |
> | 14    | 2500  | 8m 33s          | 4.8 GB          |
> | 20    | 2500  | 8m 50s          | 4.8 GB          |
>
> Using efficiency tricks (prefetching, model compilation), we reduced per-expert time from ~15 minutes to ~7 minutes for 8-task ViT-B/32. Even at 20 tasks, per-expert time remains under 9 minutes with stable memory usage.
>
> For comparison, **training-data–based adaptation methods such as Adamerging [1] require 127 minutes and 17.1 GB** in the 8 task vision setting. We thank the reviewer for highlighting this and will incorporate a compute-analysis paragraph into the main paper.

---

> > ### Author Response · Authors · 2025-11-25
> > **Continued Author Response to Reviewer R2 (FeW7) (2/2)**
> >
> > ### **R2Q1:** *“Why optimize task vectors separately? Why not a single shared vector?”*
> >
> > **AR2Q1:**
> > This is precisely what we investigate in our analysis (Table 4). We examine several strong choices for a shared vector—
> > 1. **Pretrained Model (Zero-Shot + Distill)**
> > 2. **Merged Iso-C, Iso-CTS, and TSV-M models**
> >
> > We then perform multitask distillation using auxiliary data on this shared vector.
> >
> > While this approach produces small gains (e.g., **+0.6%** for TSV-M), it remains substantially less effective than RI. In contrast, **RI + TSV-M yields +3.8%** improvement.
> >
> > We believe this gap arises from **conflicting multitask objectives** during shared-vector distillation, which cannot be effectively satisfied using task-agnostic auxiliary data.
> >
> > In contrast, RI enforces **task-specific objectives** that isolate functional subspaces—an essential requirement for robust interference reduction.
> >
> > ---
> >
> > ### **Reference**
> >
> > [1] Yang, Enneng, et al. *"Adamerging: Adaptive model merging for multi-task learning."* arXiv preprint arXiv:2310.02575 (2023).
> >
> > [2] Ilharco, Gabriel, et al. "Editing models with task arithmetic." arXiv preprint arXiv:2212.04089 (2022).
> >
> > [3] Wang, Ke, et al. "Localizing task information for improved model merging and compression." arXiv preprint arXiv:2405.07813 (2024).

---

### Official Review · Reviewer_pTFe · 2025-11-01

**Soundness:** 2
**Presentation:** 2
**Contribution:** 3
**Rating:** 4
**Confidence:** 5

**Summary:**

This paper proposes a model merging framework enhanced by a Representation Independence (RI) Loss. By leveraging auxiliary data and introducing a regularization constraint, the method aims to mitigate task interference during model merging, thereby improving robustness and generalization. The core idea is to enforce task-irrelevance constraints between merged representations, allowing the final model to better retain knowledge from each task. Experiments on multi-task benchmarks demonstrate consistent improvements.

**Strengths:**

**Clear innovation**: The RI Loss provides a novel regularization mechanism to reduce task interference, offering a fresh perspective for model merging research.

**Simple and practical**: The method can be seamlessly integrated into existing training setups as a plug-and-play module.

**Empirical validation**: Extensive evaluations across multiple benchmarks show promising gains.

Solid motivation: The task-relatedness perspective provides a sound theoretical rationale for reducing interference in merging.

**Weaknesses:**

**1，Dependency on auxiliary data**: The effectiveness appears tied to the availability and distribution of auxiliary data, yet its limitations are not thoroughly examined.

**2，Missing key baselines**: Recent strong merging methods (e.g., Wudi-Merging) are not included in comparisons.

**3，Limited large-model experiments**: The study focuses on moderate-scale models, leaving the applicability to large-scale vision-language or language models unverified.

**4，Incomplete presentation**: The main figure only explains the RI Loss without clearly visualizing the merging pipeline, which may confuse readers unfamiliar with the topic.

**5，Insufficient training-cost analysis**: More detailed benchmarking against alternatives (e.g., Adamerging, model surgery, naive distillation, multi-task learning) would strengthen the claims.

**Questions:**

1，How does RI Loss perform when the distribution of auxiliary data significantly differs from that of the target tasks? Are there failure cases or ablation studies? Does the amount of auxiliary data matter?

2，Will the authors include comparisons with recent baselines such as Wudi-Merging?

3，Can the approach scale to large models (e.g., LLMs, CLIP-based multi-task settings)? Any preliminary results or discussion?

4，Could the paper include a clearer pipeline diagram to illustrate the full merging process rather than only the RI Loss component?

5，Could the authors provide quantitative training-time comparisons with Adamerging, model surgery, naive distillation, and multi-task learning?

Minor Issues / Typos

Line 58: Incorrect citation — Task Arithmetic in the Tangent Space should be NeurIPS 2023, not 2024.

---

> ### Author Response · Authors · 2025-11-29
> **Author Response to Reviewer R1 (pTFe) (1/2)**
>
> We sincerely thank Reviewer R1 (pTFe) for the thoughtful and constructive feedback and for acknowledging the **novelty** of the RI loss, its **practical plug-and-play design**, followed by **strong motivation and experimental validation across benchmarks**. Below, we address each of the reviewer’s concerns in detail and provide additional experiments and clarifications to further strengthen the paper.
>
> ---
>
> **R1W1 & R1Q1: Dependency on Auxiliary Data**
>
> **AR1W1:** We address the reviewer’s concerns in three parts.
>
> **(a) Effectiveness of RI when auxiliary data differs from target-task data**
>
> As discussed in L424–430 and shown in Fig. 4, RI is robust to the choice of auxiliary data. Across all experiments, the auxiliary datasets are semantically unrelated to any of the 8/14/20 tasks, yet RI consistently improves merging performance. From analysis in Fig. 4, we observe that datasets with high resolution and high visual diversity (e.g., ImageNet, OxfordPets, STL10, Flowers102) yield the strongest gains.
>
> To stress-test RI, we also evaluate pure Gaussian noise as auxiliary input. While less effective than natural images, even noise improves merging performance, indicating that RI benefits from diverse visual tokens rather than semantic alignment.
>
> **(b) Ablation on Failure Cases**
>
> Conversely, failure cases naturally occur when the auxiliary data lacks visual diversity. As shown in Fig. 4, low-resolution datasets (e.g., CIFAR-100) or datasets with limited visual diversity (FER2013, PCAM) are less effective at reducing RI loss.
>
> Overall, RI does not rely on semantic similarity to tasks and performs well across a wide range of auxiliary sources, provided they offer sufficient visual complexity.
>
> **(c) Does the amount of auxiliary data matter?**
>
> We thank the reviewer for raising this important question. RI is intentionally lightweight:
> - 2500 steps × batch size 128 ≈ 320k images (≈25% of ImageNet)
>
> To examine sensitivity to dataset size, we subsample the auxiliary data to 0.8×, 0.6×, 0.4×, and 0.2× of the original 320k images. Results are presented in Table A below.
>
> **Table A: Effectiveness of RI under varying auxiliary dataset size**
> *(Data fraction 1.0 = 320k images used over 2500 steps at batch size 128)*
>
> | Method   | No RI | 1.0× | 0.8× | 0.6× | 0.4× | 0.2× |
> |----------|-------|------|------|------|------|------|
> | Iso-C    | 75.1 | 77.4 | 77.4 | 77.5 | 77.6 | 77.8 |
> | Iso-CTS  | 77.7 | 79.1 | 79.0 | 79.0 | 79.1 | 79.1 |
> | TSV-M    | 76.5 | 80.1 | 80.1 | 80.2 | 80.2 | 80.3 |
>
> We observe that even when using only **20%** of the auxiliary dataset, RI remains equally effective across SOTA merging techniques. This further confirms RI’s data efficiency and robustness to auxiliary data size.
>
> ---
>
> **R1W2 & R1Q2: Comparison with WUDI merging baseline**
>
> **AR1W2:** We appreciate the suggestion and agree that WUDI is an important baseline to include. We evaluated WUDI + RI across all 8/14/20-task setups for ViT-B/32, ViT-B/16, and ViT-L/14. The results are shown in Table B below.
>
> **Table B: Evaluating WUDI merging with RI on the vision 8/14/20-task benchmark**
>
> | Merging Method |          |   ViT-B/32     |        | |          |  ViT-B/16      |        | |          |  ViT-L/14      |        |
> |----------------|----------------------|--------|--------|-|----------------------|--------|--------|-|----------------------|--------|--------|
> |                               | 8 Tasks  | 14 Tasks | 20 Tasks | |   8 Tasks | 14 Tasks | 20 Tasks | | 8 Tasks            | 14 Tasks | 20 Tasks |
> | WUDI                      | 82.0       |      73.5    |      58.4   | |    85.2     | 77.2   | 63.8   | | 91.0               | 84.9   | 76.2   |
> | WUDI + RI (Ours)    | **84.1**  |   **77.6** |   **68.0**  | |   **86.9** | **81.6** | **74.9** | | **91.6**           | **86.3** | **82.4** |
>
> A key empirical pattern across all architectures is that merging performance degrades sharply as the number of tasks increases. This is attributed to the growing cross-task interference as the number of experts to be merged increases. RI consistently mitigates this degradation and becomes increasingly beneficial as the task count grows.
>
> Concretely, RI improves WUDI consistently: **+2.1-9.7% on ViT-B/32, +1.7-11.1% on ViT-B/16, and +0.6-6.2% on ViT-L/14, with larger gains at higher task counts**. These results confirm that RI consistently enhances even the strongest dataless merging techniques, including those explicitly designed to reduce interference. We will integrate these results into Table 1 in the main paper.
>
> The WUDI baseline accuracy (82.0%) differs from the 85.2% reported in the original WUDI paper on the 8-task ViT-B/32 setting due to the two works relying on different model checkpoints. WUDI uses checkpoints from [1], whereas our experiments use the 8/14/20-task checkpoint suite released by [2], which provides ViT-based OpenCLIP models across all task configurations.

---

> ### Author Response · Authors · 2025-11-29
> **Author Response to Reviewer R1 (pTFe) (2/2)**
>
> **R1W3 & R3Q3: Can the approach scale to large models (e.g., LLMs, CLIP-based multi-task settings)?**
>
> **AR1W3:** As noted in L207, our experiments already use CLIP-based models and evaluate RI in both in-domain and out-of-domain multi-task settings. Since RI operates solely on task-head logits and makes no modality- or architecture-specific assumptions, the method should naturally extend to large Transformer-based vision–language models and LLMs. We will add this discussion to the main paper.
>
> We agree that evaluation on larger models would strengthen the work. While LLM-scale experiments are beyond our current scope, the strong results on CLIP models and the architecture-agnostic design suggest promising scalability. We view LLM-scale validation as important future work.
>
> ---
>
> **R1W4 & R1Q4: Request for a full pipeline diagram**
>
> **AR1W4 & AR1Q4:** We thank the reviewer for this helpful suggestion. We agree that a visual illustration would improve clarity, especially for readers new to model merging. We will include a complete end-to-end diagram showcasing:
>
>  (1) expert models → (2) RI adaptation → (3) merged vector
>
> ---
>
> **R1W5 & R1Q5: Benchmarking against alternatives such as Adamerging, Model Surgery, naive Distillation, and Multi-Task Learning**
>
> **AR1W5 & AR1Q5:** We appreciate this question. However, the proposed approaches operate in a different setting. These adaptation methods **require access to task-specific data**, which is fundamentally incompatible with the **task-data–free setting** that RI is designed for.
> RI’s objective explicitly enforces that expert models become functionally orthogonal using just auxiliary unlabeled data, making it applicable in scenarios where original task data is unavailable.
>
> ---
>
> **Minor Issues / Typos**
>
> We thank the reviewer for catching the citation error on L58. Ortiz-Jimenez et al. (*Task Arithmetic in the Tangent Space*) is NeurIPS 2023, not 2024. We will correct this.
>
> ---
>
> To conclude, the new experiments strengthen our claims by: **(1) Demonstrating robustness to 80% reduction in auxiliary data and (2) Showing RI improves even state-of-the-art WUDI by up to 11.1%**. We thank the reviewer for the thorough and constructive feedback, which has significantly strengthened our paper. We believe the revised version addresses all major concerns.
>
> ---
>
> **References:**
>
> [1] Ilharco, Gabriel, et al. "Editing models with task arithmetic." arXiv preprint arXiv:2212.04089 (2022).
>
> [2] Wang, Ke, et al. "Localizing task information for improved model merging and compression." arXiv preprint arXiv:2405.07813 (2024).

---

### Meta-Review · Area_Chair_aAV8 · 2026-01-02

**Summary:**

This paper proposes a new model merging method that improves merging performance by addressing cross-task interference. However, the proposed approach requires gradient-based optimization for each expert, which incurs high computational costs in scenarios with large model sizes or many tasks. In addition, the set of baseline model merging methods used for comparison is neither comprehensive nor up to date, with several relevant works missing. Finally, the evaluation is limited to image classification tasks, lacking validation on large language models and text generation tasks.

**Reviewer Concerns:**

All four reviewers point out that the paper is evaluated only on simple image classification tasks, lacking effective validation on large language models and text generation tasks. In addition, Reviewer 1att notes that the proposed method relies on expensive optimization, requiring multiple steps of gradient descent for each expert model, which makes it difficult to scale to a large number of tasks. Finally, both Reviewer pTFe and Reviewer 1att point out that the paper is missing comparisons with many relevant baselines.

**Reviewer Scores:**

Reviewer pTFe suggests comparing against the latest baseline, WUDI-Merging, and the authors added this comparison during the rebuttal. However, issues regarding the reliance of the proposed method on auxiliary data, the lack of validation on large language models, and the absence of comparisons with alternative approaches (e.g., AdaMerging, model surgery, naive distillation, and multi-task learning) were not well addressed during the rebuttal.

Reviewer FeW7 is mainly concerned about how the performance gains are presented and the comparison with WUDI-Merging. The authors provided explanations and additional comparisons during the rebuttal. However, the effectiveness of the proposed method on large models was still not adequately addressed.

Reviewer 1att’s main concerns are the lack of comparisons with more recent methods, such as Surgery, TwinMerging, CatMerging, and LoTMerging, which were not included during the rebuttal. In addition, the reviewer is concerned about the efficiency of the proposed method and its effectiveness on large language models, since it requires optimizing each expert model and is evaluated only on classification tasks, without discussion of generation tasks or large models. These shortcomings were not resolved during the rebuttal.

Reviewer SjwS finds the contributions of the paper unclear and notes that some results lack sufficient explanation. The authors provided clarifications during the rebuttal. However, no additional experiments were added to demonstrate the effectiveness of the proposed method on text generation tasks during the rebuttal.

---

### Decision · Program_Chairs · 2026-01-26

Reject